# Determination of Referential Rates for Optical Coherence Tomography and Optical Coherence Tomography Angiography Flow Deficits in the Macular Choriocapillaris in Ophthalmologically Healthy Children

**DOI:** 10.3390/medicina56050238

**Published:** 2020-05-16

**Authors:** Viktorija Bakstytė, Liveta Šniurevičiūtė, Evelina Šimienė, Justina Skruodytė, Ingrida Janulevičienė

**Affiliations:** 1Faculty of Medicine, Medical Academy, Lithuanian University of Health Sciences, LT-44307 Kaunas, Lithuania; viktorija.bakstyte@gmail.com; 2Department of Ophthalmology, Hospital of Lithuanian University of Health Sciences Kaunas Clinics, LT-50162 Kaunas, Lithuania; evelina.maciulaityte@gmail.com (E.Š.); skruodytes@gmail.com (J.S.); Ingrida.Januleviciene@kaunoklinikos.lt (I.J.)

**Keywords:** ophthalmologically healthy children, retinal nerve fiber layer, choriocapillaris, swept-source optical coherence tomography, optical coherence tomography angiography, flow deficits

## Abstract

*Background and Objectives*: Despite the growing number of new research publications, normative references for children’s optical coherence tomography (OCT) parameters are still not completed. We chose to explore this topic because of the lack of normative parameters that is due to an improvement in different technologies and instruments. Our aim was to determine referential rates of retinal nerve fiber layer (RNFL) thickness and flow deficits (FD%) in the macular choriocapillaris (CC) in normal eyes of ophthalmologically healthy children. *Materials and Methods*: Ophthalmologically healthy 8- to 14-year-old individuals participated (*n* = 75) in this study. OCT images were taken using an swept-source-OCT (SS-OCT) instrument (DRI-OCT Triton, Topcon, Tokyo, Japan). The early treatment diabetic retinopathy study (EDTRS) grid (6 × 6 mm) divided the RNFL into the thickness maps. The FD% values of the CC were calculated on the 3 × 3-mm scans in a 1-mm circle (C_1_), 1.5-mm rim (R_1.5_), and the entire 2.5-mm circle (C_2.5_), and on the 6 x 6-mm scans in a 1-mm circle (C_1_), 1.5-mm rim (R_1.5_), the entire 2.5-mm circle (C_2.5_), 2.5-mm rim (R_2.5_), and 5-mm circle (C_5_). *Results*: Both scan quantifications of FD% in the C_1_, C_2.5_, and R_1.5_ sectors were similar, but the 6 × 6-mm scan measurements were statistically significantly smaller than the 3 × 3-mm ones. Significant moderate correlations were found between axial length (AxL) and FD% in the 6 × 6-mm scans, namely C_1_ (*r* = −0.347, *p* = 0.002), C_2.5_ (*r* = −0.337, *p* = 0.003), R_1.5_ (*r* = −0.328, *p* = 0.004), R_2.5_ (*r* = −0.306, *p* = 0.008), and C_5_ (*r* = −0.314, *p* = 0.006). *Conclusions*: The thinnest RNFL layers were on the temporal and nasal sides. FD% values in the C_1_, C_2.5_, and R_1.5_ sectors were similar between the 3 × 3-mm and 6 × 6-mm scans. The negative moderate correlations between AxL and FD% were found in all C_1_, C_2.5_, C_5,_ R_1.5_, and R_2.5_ sectors of the 6 × 6-mm scans. Further prospective studies are needed to determine more accurate normative references for children’s OCT parameters.

## 1. Introduction

Since the introduction of optical coherence tomography (OCT), numerous advancements in ophthalmology have been achieved. This method creates cross-sectional imaging of the retina and choroid [1,2]. OCT is a minimally invasive investigation that visualizes microstructures of tissue without a radiation dose in real time and in vivo [1,3]. It is based on low-coherence light, which is reflected to construct a depth profile of a sample [4]. Optical coherence tomography angiography (OCTA) is a new imaging modality, developed for a three-dimensional vascular mapping of the retinal microvascularization of the fundus [5]. This method precisely detects the motion of blood cells in chorioretinal blood vessels [4,6,7].

The retinal nerve fiber layer (RNFL) is composed of the unmyelinated ganglion cell axons that converge as the optic nerve [8,9]. The RNFL is important for the detection and monitoring of congenital glaucoma, optic neuritis, fibromyalgia, optic nerve hypoplasia, and autosomal dominant optic atrophy [9,10,11]. Swept-source-OCT (SS-OCT) has the ability to measure average and sectoral RNFL thickness [11,12].

The choriocapillaris (CC) is a thin, single layer of the capillary plexus located between the Bruch’s membrane and Sattler’s layer, which gives nourishment and supplies oxygen to the retinal pigment epithelium (RPE) and outer retina [7,12]. According to clinical and histopathological studies, there are different retinal diseases, such as juvenile macular degeneration (JMD) (most commonly known as Stargardt disease) and diabetic retinopathy (DR), which correlates with the CC circulation [7,13,14,15]. Nevertheless, it is a challenge to take images of the choriocapillaris in vivo using existing technology [16]. For a long time, the standard for visualization of the CC was dye-based angiography, but it has limited depth resolution especially in deeper vascular layers [17]. Fluorescein angiography (FA) or indocyanine green angiography (ICGA) cannot be informative, because of the small intercapillary spaces in the CC’s location and the fenestrated nature of the CC [16,18]. OCTA is the main method to visualize the CC in vivo [19,20,21].

The speed and image processing capability of OCT have increased the use of the technique, especially in children [22]. Although OCT is a well-tolerated imaging modality, it has limitations, such as variations in height and position of headrests, the time taken for the scans, and the machine’s high sensitivity to movements, which result in image artefacts [1,21]. In pediatric populations, OCT has fallen behind compared to its use in adult populations [1,20]. There are many modern devices, but it is often difficult to evaluate the results for children when the normative references are determined for adults. In order to exploit the potential of OCT for the diagnosis of pediatric eye diseases, it is necessary to know the referential rates in children. Despite the growing number of new research publications, normative references for children’s OCT parameters are still not completed. We chose to explore this topic because of the lack of normative parameters that is due to an improvement in different technologies and instruments. Consequently, we decided to see if there were any significant differences in OCT normative references related to age, because it is not possible to adapt the referential rates to children and assess pathological changes [23,24].

## 2. Materials and Methods

This prospective cross-sectional study was performed at the Department of Ophthalmology of the Hospital of the Lithuanian University of Health Sciences, Kaunas Clinics between January and March 2018. In all, 75 healthy children undergoing a routine prophylactic health examination in primary school were invited to take part in this study. The Ethics Committee approved the study (BEC-LSMU(R)-02) on 15 September 2017, and the research followed the tenets of the Declaration of Helsinki. Parents or guardians approved the participation in the study and signed informed consent forms before taking part in the research. All children gave oral consent before the examination. Inclusion criteria were as follows: 8–14 years of age, no previously recorded systemic or eye diseases except for mild refractive errors (hyperopia, astigmatism up to 3D; myopia up to 6D). Exclusion criteria: not willing to attend the study, not able to take part in ophthalmic examinations, systemic diseases as recorded in medical documents by family doctor, and high refractive errors (hyperopia, astigmatism ≥3D; myopia ≥6D). All the children underwent the following examinations: best corrected visual acuity Snellen chart (VA), biomicroscopy, ophthalmoscopy. A biometer (OcuScanRxP, Irvine, CA, USA) was used to measure axial length (AxL), an autorefractometer (TONOREF™ III Nidek, Gamagori, Aichi, Japan) to obtain refractive index (SE) values, a tonometer (Icare ic200, Vantaa, Finland) to identify intraocular pressure (IOP), a pachymeter (Alcon OcuScanRxP, Irvine, CA, USA) to detect central corneal thickness (CCT), after cycloplegia OCT images were captured using an SS-OCT instrument (DRI-OCT Triton, Topcon, Tokyo, Japan). We used automated segmentation and applied manual correction, if necessary. The Topcon device defines the CC slab as 0–10.4 µm below Bruch’s membrane. The early treatment diabetic retinopathy study (EDTRS) grid (6 × 6 mm) was used to divide the RNFL into the thickness maps (µm) to gain the values for each sector. The grid was composed of four quadrants (superior, inferior, nasal, temporal) that were distributed in twelve clock-hour independent sectors around the disc.

### 2.1. Quantitative Image Analysis of the Choriocapillaris OCTA Images

OCT was performed after cycloplegia, and the patients had to sit still and focus on the study. Based on the characteristics of the OCTA manufacturers, images with a quality higher than 40 were considered suitable. Poor quality images were not included in the study. The scan protocols for imaging the CC included a 3 × 3-mm scan and a 6 × 6-mm scan that were centered on the fovea. Image sizes were 320 × 320 pixels. Each eye was captured separately, but only one eye was chosen to be analyzed in the research according to the quality of the scan. The flow deficits (FDs) of the choriocapillaris OCTA images were analyzed using ImageJ v1.43r software (National Institutes of Health, Bethesda, MD, USA). The percentages of FD (FD%) of the CC were calculated in 1-mm circle (C_1_), 1.5-mm rim (R_1.5_), and the entire 2.5-mm circle (C_2.5_) on the 3 × 3-mm scans (Figure 1). On the 6 × 6-mm scans, FD% values were quantified in a 1-mm circle (C_1_), 1.5-mm rim (R_1.5_), the entire 2.5-mm circle (C_2.5_), 2.5-mm rim (R_2.5_), and 5-mm circle (C_5_) (Figure 2). Both 3 × 3-mm and 6 × 6-mm scans were centered manually on the fovea. The Phansalkar threshold (radius, 15 pixels) was used to binarize the CC image and then “Analyze particles” was applied to count FD%. Separated parts of the grid were measured three times and then the mean value was counted to ensure accuracy.

### 2.2. Statistical Analysis

Statistical analyses were performed using the MS Excel 2010 program and the IBM Statistical Package for the Social Sciences (SPSS) software version 25 (IBM Corporation, Armonk, NY, USA). We applied the Shapiro–Wilk test and found that data were normally distributed. Descriptive statistics was used to calculate the mean values and standard deviation (SD). Differences between the 3 × 3-mm and 6 × 6-mm grids were identified using the paired samples *t*-test. Pearson’s two-tailed correlation test was used to determine the strength of correlations between the variables. Results with a *p*-value less than 0.05 (*p* < 0.05) were considered to be statistically significant. The correlation was assessed as weak, when *r* was ≤ 0.3, medium, when 0.3 < *r* ≤ 0.75, and high, when 0.75 < *r* ≤ 1.

## 3. Results

A total of 75 individuals (43 girls and 32 boys) were included in the study. The mean age of girls was 10.48 years (SD = 1.77 years) and the mean age of boys was 10.69 years (SD = 1.82 years) (*p* > 0.05; *t*-test). Analyzed eye parameters: the mean AxL was 23.02 mm (SD = 0.62 mm), the mean IOP was 15.05 mmHg (SD = 2.02 mmHg), the mean VA was 0.99 (SD = 0.06), the M SE before cycloplegia was −0.35 D (SD = 0.87 D), and the mean SE after cycloplegia was 0.59 D (SD = 0.75 D). The mean CCT was 551.59 microns (SD = 34.31 microns).The results of minimum, maximum, mean, and SD of the RNFL are shown in Table 1.

Insignificant correlations were found between age of participants and RNFL inferior (*r* = −0.113, *p* = 0.351), RNFL nasal (*r* = 0.008, *p* = 0.950), RNFL temporal (*r* = −0.044, *p* = 0.719), and RNFL superior (*r* = −0.112, *p* = 0.356).

We investigated insignificant associations between AxL and RNFL inferior (*r* = −0.147, *p* = 0.211), RNFL nasal (*r* = −0.176, *p* = 0.133), RNFL temporal (*r* = 0.215 *p* = 0.066), and RNFL superior (*r* = 0.028, *p* = 0.812).

From both scans, the quantifications of FD% in C_1_, C_2.5_, and R_1.5_ sectors were comparable (*p* = 0.0001) (Figure 3). The 6 × 6-mm scan measurements were statistically significantly smaller than the 3 × 3-mm scan measurements (Figure 4).

Significant moderate correlations were found between AxL and FD% in the 6 × 6-mm scans C_1_ (*r* = −0.347, *p* = 0.002), C_2.5_ (*r* = −0.337, *p* = 0.003), R_1.5_ (*r* = −0.328, *p* = 0.004), R_2.5_ (*r* = −0.306, *p* = 0.008), and C_5_ (*r* = −0.314, *p* = 0.006). However, associations between AxL and FD% in the 3 × 3-mm scans C_1_ (*r* = −0.129, *p* = 0.269), C_2.5_ (*r* = −0.049, *p* = 0.675), and R_1.5_ (*r* = −0.009, *p* = 0.942) were insignificant.

There were insignificant correlations between SE before cycloplegia and FD% in the 3 × 3-mm C_1_ (*r* = −0.119, *p* = 0.312), C_2.5_ (*r* = −0.101, *p* = 0.392), and R_1.5_ (*r* = −0.103, *p* = 0.385) and in the 6 × 6-mm C_1_ (*r* = 0.118, *p* = 0.321), C_2.5_ (*r* = 0.050, *p* = 0.672), R_1.5_ (*r* = 0.046, *p* = 0.698), R_2.5_ (*r* = 0.101, *p* = 0.395), and C_5_ (*r* = 0.050, *p* = 0.675) scans.

Age of participants was insignificantly associated with FD% in the 3 × 3-mm C_1_ (*r* = 0.032, *p* = 0.791), C_2.5_ (*r* = −0.175, *p* = 0.145), and R_1.5_ (*r* = −0.214, *p* = 0.074) and in the 6 × 6-mm C_1_ (*r* = 0.109, *p* = 0.369), C_2.5_ (*r* = 0.023, *p* = 0.850), R_1.5_ (*r* = −0.012, *p* = 0.925), R_2.5_ (*r* = −0.098, *p* = 0.419), C_5_ (*r* = −0.048, *p* = 0.693) scans.

## 4. Discussion

In the present study, the aim was to determine normative references for children’s OCT and OCTA parameters. Our results showed that the thinnest RNFLs were temporal (namely RNFL 2, RNFL 3, and RNFL 4) and nasal (namely RNFL 8, RNFL 9, and RNFL 10) and the thickest layers were superior (namely RNFL 11, RNFL 12, and RNFL 1) and inferior (namely RNFL 5, RNFL 6, and RNFL 7). L. Devang et al. also found that the RNFL in the temporal quadrant was the thinnest and identified the inferior quadrant as the thickest in ophthalmologically healthy children [25]. F. Gürağaç et al. carried out an investigation, where they examined healthy Turkish children aged between 3 and 17 years and presented the same research results: the thickest quadrant was inferior and the thinnest one was temporal [26]. Z. Yang et al. obtained the same results in healthy adults, except the thickest layer of the RNFL was superior [13]. El-Dairi et al. indicated that the RNFL is race dependent and black children have higher RNFL thickness values than white children, especially their superior quadrant was greater [27]. In our research, a relationship between age of participants, AxL, and RNFL was detected, but it was not statistically significant. Compared to the investigation of M. Cubuk et al., no statistically significant correlations were also found between age and RNFL in ophthalmologically healthy adults [28]. In our study, we observed that FD% values in the C_1_, C_2.5_, and R_1.5_ sectors were similar between the 3 × 3-mm and 6 × 6-mm scans. The measurements of FD% in the 3 × 3-mm scans were larger than those in the 6 × 6-mm, because the 3 × 3-mm scans were more accurate and included much finer details. Therefore, the 3 × 3-mm scans increased the sensitivity of the measurements, but the examined area of the scans was smaller than on the 6 × 6-mm scans. The FD values we obtained were similar to the normative FD rates in adults. There are studies that mention an increase in FD before myopia is diagnosed [16]. We found insignificant correlations between AxL and FD% in the 3 × 3-mm scans. The negative moderate correlations between AxL and FD% were found in all C_1_, C_2.5_, C_5_, R_1.5_, and R_2.5_ sectors of the 6 × 6-mm scans. However, in their study, F. Zheng et al. determined that there were no meaningful correlations between FD% and axial length (|r| < 0.30) in 20- to 80-year-old healthy adults [29]. Some correlations were found between SE before cycloplegia and FD% in the C_1_, C_2.5_, and R_1.5_ sectors of the 3 × 3-mm scans and in the C_1_, C_2.5_, C_5_, R_1.5_, and R_2.5_ sectors of the 6 × 6-mm scans, but these correlations were not statistically significant. Moreover, no statistically significant correlations were found between age of participants and FD% in the C_1_, C_2.5_, and R_1.5_ sectors of the 3 × 3-mm scans and in the C_1_, C_2.5_, C_5_, R_1.5_, and R_2.5_ sectors of the 6 × 6-mm scans. Our results would be helpful for the diagnosis of retinal diseases, especially JMD and DR in children [14,30]. However, future studies are needed to analyze whether CC FD% correlates with other pathologies in JMD, such as drusen and macular neovascularization in children’s eyes [31].

We faced several difficulties while performing this study that could lead to drawbacks that we need to mention. It is always challenging to perform studies in pediatric patients, because of the stricter requirements from the Ethics Committee and Institutional Review Board. It was not easy to obtain the sufficient sample, because many children already had refractive errors and high myopia, which was an exclusion criterion in our study. It was also difficult to obtain the consent from parents or guardians required for participation in the study as most of them were busy and had no time for additional investigations. Additionally, many children did not follow instructions correctly and some examinations were considered as statistically non reliable. Another limitation of our work was that we could not consider diurnal variations of CC perfusion [32]. Tan et al. studied how circadian rhythms affect subfoveal choroidal thickness and they found that the highest values were in the morning at 9:00 a.m. and the lowest values were in the evening at 5:00 p.m. [33]. Therefore, further prospective studies are needed to set more accurate normative references for children’s OCT parameters.

## 5. Conclusions

In this study, we determined referential rates for SS-OCT RNFL thickness and OCTA FD% in the macular CC in ophthalmologically healthy children. The thinnest RNFL layers were on the temporal and nasal sides. FD% values in the C_1_, C_2.5_, and R_1.5_ sectors were similar between the 3 × 3-mm and 6 × 6-mm scans. The negative moderate correlations between AxL and FD% were found in all 1-mm circle, 1.5-mm rim, 2.5-mm circle, 2.5-mm rim, and 5-mm circle sectors of the 6 × 6-mm scans.

These results will be useful for future research and diagnostic interpretation in pediatric ophthalmological patients. Further prospective studies of different age groups and follow-up over time are needed to determine more accurate normative references for children’s RNFL and OCT angiography parameters.

## Figures and Tables

**Figure 1 medicina-56-00238-f001:**
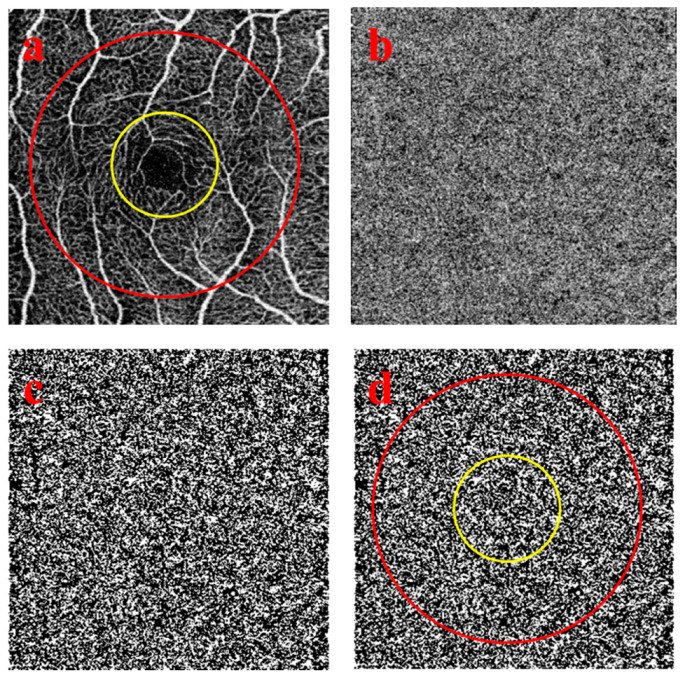
3 × 3-mm-diameter scans used for visualization of the choriocapillaris and the areas used to calculate the percentage of flow deficits (FDs). (**a**) shows the 1-mm-diameter yellow circle and 2.5-mm-diameter red circle centered on the fovea that includes 3 regions: the 2.5-mm circle composed of a 1-mm circle and a 1.5-mm rim. (**b**) represents the choriocapillaris layer after artefact removal. (**c**) presents the choriocapillaris layer after binarization and thresholding. The white areas in the scan correspond to the flow deficits. (**d**) shows the circles used to measure the percentage of flow deficits within the regions. (**a**–**d**) Images from the same 9-year-old subject.

**Figure 2 medicina-56-00238-f002:**
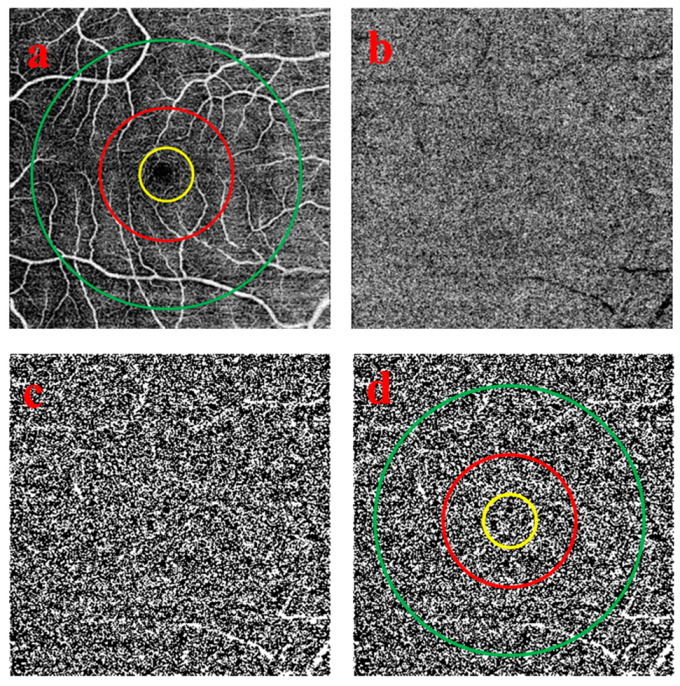
6 × 6-mm-diameter scans used for visualization of the choriocapillaris and the areas used to calculate the percentage of flow deficits. (**a**) shows the 1-mm-diameter yellow circle, 2.5-mm-diameter red circle, and 5-mm-diameter green circle centered on the fovea that includes 4 regions: the 5-mm circle composed of a 1-mm circle, a 1.5-mm inner rim, and a 2.5-mm outer rim. (**b**) represents the choriocapillaris layer after artefact removal. (**c**) presents the choriocapillaris layer after binarization and thresholding. The white areas in the scan correspond to the flow deficits. (**d**) shows the circles used to measure the percentage of flow deficits within the regions. (**a**–**d**) Images from the same 9-year-old subject.

**Figure 3 medicina-56-00238-f003:**
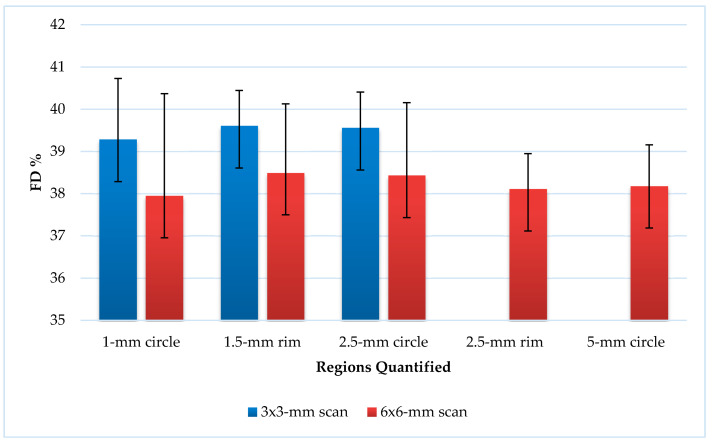
The percentage of flow deficits in the 3 × 3-mm and 6 × 6-mm scans.

**Figure 4 medicina-56-00238-f004:**
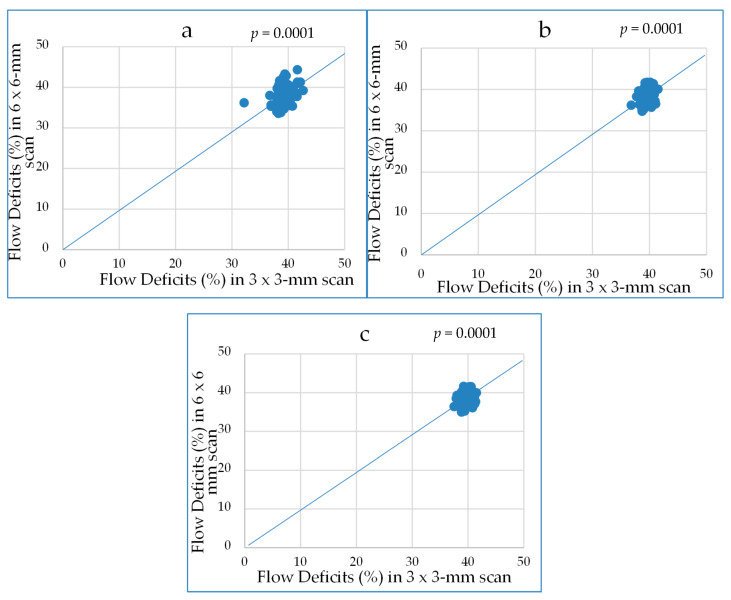
The percentage of flow deficits in the (**a**) 1-mm circles, (**b**) 2.5-mm circles, and (**c**) 1.5-mm rims centered on fovea and compared between 3 × 3-mm and 6 × 6-mm scans.

**Table 1 medicina-56-00238-t001:** Minimum, maximum, mean, and standard deviations of the retinal nerve fiber layer retinal nerve fiber layer (RNFL).

RNFL	Minimum	Maximum	Mean	Standard Deviation
RNFL 1	100	190	147.89	21.469
RNFL 2	63	166	89.88	16.991
RNFL 3	48	98	63.95	8.935
RNFL 4	51	128	76.93	14.693
RNFL 5	108	196	149.23	19.888
RNFL 6	82	204	146.88	24.644
RNFL 7	71	161	108.35	19.814
RNFL 8	45	121	80.68	15.197
RNFL 9	46	95	63.78	10.852
RNFL 10	69	145	100.22	17.084
RNFL 11	86	163	123.66	18.763
RNFL 12	65	183	121.66	24.826
RNFL inferior	99	169	134.85	15.459
RNFL nasal	54	115	81.62	11.932
RNFL temporal	56	128	76.95	11.825
RNFL superior	96	166	131.04	14.126

Distribution of RNFL quadrants (superior, inferior, nasal, temporal) and twelve clock-hour (RNFL 1–12) independent sectors around the disc.

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
