# Peer review of "Determination of Referential Rates for Optical Coherence Tomography and Optical Coherence Tomography Angiography Flow Deficits in the Macular Choriocapillaris in Ophthalmologically Healthy Children"

_medicina, 2020, doi:10.3390/medicina56050238_

Round 1
Reviewer 1 Report
Bakstyte et al. present a study that deals with RNFL thickness measurements using OCT, and choriocapillaris flow using OCTA in healthy children. Both techniques are fast, easy to obtain, and non-invasive, which make them important tools in ophthalmological examination and follow-up studies including diagnostic in children. Gaining referential data is of significant importance for interpretation of the diagnostic and for future research, since there are only few studies dealing with OCT in healthy children.
In this work, RNFL thickness according to the EDTRS grid was obtained in 75 children, decreasing from inferior to superior to nasal and to temporal. The choriocapillaris flow was similar between 3x3mm and 6x6mm scans for C1, C2.5 and R1.5 sectors. Figures are of good quality and legends are intelligible. The number of subjects is great.
The following issues need to be addressed:
- The number of subjects (n=75) should be mentioned in the abstract, because it is a significant strength of this study. When using abbreviations in the abstract, they need to be defined (AxL in line 25).
- The introduction remains quite superficial. It should highlight why the study is of importance and should define the purpose and its significance more specific.
- The location of the CC should be revised. It is located between Bruch’s membrane and Sattler’s layer (p. 2, line 49).
- Materials and Methods are well written and contain the important details. The Phansalkar threshold was used for binarization, which has been proven to be sufficient. However, the segmentation parameters of the CC should be mentioned. Furthermore, it should be stated if OCT/OCTA images were taken after or before cycloplegia.
- The major results are presented. The authors used the T-test to identify differences between the 3x3 and 6x6 mm images. However, p-values should be reported (p. 5, lines 144-146).
- The discussion needs to be revised. The statements should be checked and interpreted more carefully (e.g. “We found no statistically significant, but weak correlations …” (p. 7, l. 177-178). Furthermore, the results were barely discussed in perspective of existing studies. Especially regarding RNFL thickness there are lots of studies, whose discussion would improve the quality of this work. Overall, the focus of the discussion should be on the OCTA results, since these are the main novelty of the study.
- In OCTA, different imaging artifacts can be observed that can strongly influence quantitative analysis. Did you check the quality of automated segmentation in all images? How was motion, projection or segmentation artifacts dealt with? Please consider: Lauermann JL et al., Prevalences of segmentation errors and motion artifacts in OCT-angiography differ among retinal diseases. Graefes Arch Clin Exp Ophthalmol. 2018
- In adults, choroidal perfusion values are known to underlay physiological diurnal variation. Please state, whether OCTA images were obtained at a similar time of the day. Otherwise, please mention as a limitation. Consider: Siegfried F et al., Evaluating diurnal changes in choroidal sublayer perfusion using optical coherence tomography angiography. ACTA Ophthalmol. 2019
Sarwar S et al., Diurnal variation of choriocapillaris vessel flow density in normal subjects measured using optical coherence tomography angiography. Int J Retina Vitreous. 2018 - There are some typos and grammatical issues throughout the manuscript, which should be edited.
Author Response
Point 1: The number of subjects (n=75) should be mentioned in the abstract, because it is a significant strength of this study. When using abbreviations in the abstract, they need to be defined (AxL in line 25).
Response 1: Thank you for pointing this out. We mentioned the number of subjects (n=75) in line 21 and defined AxL in line 28.
Point 2: The introduction remains quite superficial. It should highlight why the study is of importance and should define the purpose and its significance more specific.
Response 2: We added to the introduction: In order to exploit the potential of OCT for the diagnosis of pediatric eye diseases, it is necessary to know the referential rates in children. Changes are made in lines 68-70.
Point 3: The location of the CC should be revised. It is located between Bruch’s membrane and Sattler’s layer (p. 2, line 49).
Response 3: Thank you for your comment. We changed the CC location based on your comments in line 53.
Point 4: Materials and Methods are well written and contain the important details. The Phansalkar threshold was used for binarization, which has been proven to be sufficient. However, the segmentation parameters of the CC should be mentioned. Furthermore, it should be stated if OCT/OCTA images were taken after or before cycloplegia.
Response 4: We added to the Materials and Methods: OCT/OCTA images were taken after cycloplegia. An experienced grader (JCW) was used to get automated segmentation and it was manually corrected if necessary. Changes are made in lines 92-95 and 99.
Point 5: The major results are presented. The authors used the T-test to identify differences between the 3x3 and 6x6 mm images. However, p-values should be reported (p. 5, lines 144-146).
Response 5: Thank you for your suggestion. We mentioned p-values (p=0.0001) in the figure 4 and in 157 line.
Point 6: The discussion needs to be revised. The statements should be checked and interpreted more carefully (e.g. “We found no statistically significant, but weak correlations …” (p. 7, l. 177-178). Furthermore, the results were barely discussed in perspective of existing studies. Especially regarding RNFL thickness there are lots of studies, whose discussion would improve the quality of this work. Overall, the focus of the discussion should be on the OCTA results, since these are the main novelty of the study.
Response 6: We have slightly modified and supplemented the discussion: L. Devang et al. also found that RNFL in the temporal quadrant is the thinnest and identified inferior quadrant as the thickest in ophthalmologically healthy children. [25]. F. Gürağaç et al. carried out an investigation, where they examined healthy Turkish children aged between 3 and 17 years and presented the same research results: the thickiest quadrant was inferior and the thinnest one was temporal [26]. El-Dairi et al. indicated that RNFL is race dependent and black children have higher RNFL thickness values than white children, especially their superior quadrant was greater [27].
As limitation of our work we mentioned that we could not consider diurnal variations of CC perfusion [32]. We found that Tan et al. studied how circadian rhythms affect subfoveal choroidal thickness and they found that the highest values were in the morning at 9:00 and the lowest values were in the evening 5:00 [33]
Changes are made in lines 190-198, 210-212 and 227-230.
Point 7: In OCTA, different imaging artifacts can be observed that can strongly influence quantitative analysis. Did you check the quality of automated segmentation in all images? How was motion, projection or segmentation artifacts dealt with? Please consider: Lauermann JL et al., Prevalences of segmentation errors and motion artifacts in OCT-angiography differ among retinal diseases. Graefes Arch Clin Exp Ophthalmol. 2018
Response 7: In order to avoid artifacts an experienced grader (JCW) was used to get automated segmentation and it was manually corrected if necessary. OCT was performed after cycloplegia, patients had to sit still and focus on the study. Based on the characteristics of the OCTA manufacturers, images with a quality higher than 40 were considered suitable. This is mentioned in lines 93-95 and 99-101.
Point 8: In adults, choroidal perfusion values are known to underlay physiological diurnal variation. Please state, whether OCTA images were obtained at a similar time of the day. Otherwise, please mention as a limitation. Consider: Siegfried F et al., Evaluating diurnal changes in choroidal sublayer perfusion using optical coherence tomography angiography. ACTA Ophthalmol. 2019 Sarwar S et al., Diurnal variation of choriocapillaris vessel flow density in normal subjects measured using optical coherence tomography angiography. Int J Retina Vitreous. 2018
Response 8: We added to the discussion: Another limitation of our work is that we could not consider diurnal variations of CC perfusion [29]. Tan et al. studied how circadian rhythms affect subfoveal choridal thickness and they found that the highest values were in the morning at 9:00 and the lowest values were in the evening 5:00 [30]. Changes are made in lines 227-230.
Point 9: There are some typos and grammatical issues throughout the manuscript, which should be edited.
Response 9: Thank you for your comment. We tried to review all grammatical mistakes.
Reviewer 2 Report
- This study aimed to evaluate the referential rates of RNFL and FD of CC in OCTA. Do the authors believe the number of participants was enough to determine the normative references? So, what is the exact normal range of these parameters in the children?
- The authors asserted that they have found weak correlation or association between age or AxL and RNFL thickness (Line 142-147). However, this reviewer could not find any correlation or association. The authors should provide the exact standard for assessing the statistical significance, for example, weak or moderate correlation in the method section.
- What is the meaning of the number (1 to 12) that was described after RNFL in the Table 1?
- Please provide evidences showing that FD% is smaller in the 6x6 mm than the 3x3 mm scan. (Line 149-150). Figure 3 does not support the authors’ description. The statistical significance could be described as a p value.
- The contents in Table 2 would be better to be depicted as a bar graph or a box-and-whisker plot, because the graph can show the statistical significance effectively at a glance.
- The discussion section did not fully give an insight into the significance of this study. The authors need to discuss in depth about the major findings of this study. In addition, the authors could emphasize the significance and limitation of this study, especially for the study in pediatric patients, not only about getting a consent but also about the content itself that was different from the adult. How many kids were excluded due to non-reliable examinations?
Author Response
Point 1: This study aimed to evaluate the referential rates of RNFL and FD of CC in OCTA. Do the authors believe the number of participants was enough to determine the normative references? So, what is the exact normal range of these parameters in the children?
Response 1: We agree, that further studies with different age groups will be more reliable, but now we think that this study would be useful for future research and diagnostic interpretation in pediatric ophthalmological patients, since there are only few studies dealing with OCTA in healthy children. Moreover, OCT and OCTA are fast, easy to obtain, and non-invasive, which make them important tools in ophthalmological examination and follow-up studies including diagnostic in children.
Point 2: The authors asserted that they have found weak correlation or association between age or AxL and RNFL thickness (Line 142-147). However, this reviewer could not find any correlation or association. The authors should provide the exact standard for assessing the statistical significance, for example, weak or moderate correlation in the method section.
Response 2: Thank you for pointing this out. In materials and methods we added that the correlation was assessed as weak, when r was ≤ 0.3, medium, when 0.3 < r ≤ 0.75, and high, when 0.75 < r ≤ 1 (changes are made in lines 136-137).We also corrected correlation’s power without significance as accidental coincidence (changes are made in lines 150-153; 175; 176; 180; 198; 200; 207; 211 and 213).
Point 3: What is the meaning of the number (1 to 12) that was described after RNFL in the Table 1?
Response 3: RNFL‘s grid was composed of four quadrants (superior, inferior, nasal, temporal) that were distributed in twelve clock-hour independent sectors around the disc. We wrote about that in materials and methods and we also added in the Table 1 (changes made in lines 148-149).
Point 4: Please provide evidences showing that FD% is smaller in the 6x6 mm than the 3x3 mm scan. (Line 149-150). Figure 3 does not support the authors’ description. The statistical significance could be described as a p value.
Response 4: Thank you Reviewer for your comment. We added the statistical significance as a p value in the figure 4 (changes are made in lines 166-168) and accordingly in the text (changes are made in line 156).
Point 5: The contents in Table 2 would be better to be depicted as a bar graph or a box-and-whisker plot, because the graph can show the statistical significance effectively at a glance.
Response 5: Thank you for your suggestion. We changed Table 2 in a bar graph (changes are made in lines 163-164), but in Table 2 we wanted to show minimum, maximum values and the percentage of flow deficits in the 3x3-mm and 6x6-mm scans. We revealed the percentage of flow deficits in the 1 mm circles, 2.5-mm circles, 1.5-mm rims centered on fovea and compared between 3x3-mm and 6x6-mm scans with statistical significance (figures are in lines 166-168).
Point 6: The discussion section did not fully give an insight into the significance of this study. The authors need to discuss in depth about the major findings of this study. In addition, the authors could emphasize the significance and limitation of this study, especially for the study in pediatric patients, not only about getting a consent but also about the content itself that was different from the adult. How many kids were excluded due to non-reliable examinations?
Response 6: We added some insights into Discussion showing the importance of pediatric patients examinations. L. Devang et al. also found that RNFL in the temporal quadrant is the thinnest and identified inferior quadrant as the thickest in ophthalmologically healthy children. [25]. F. Gürağaç et al. carried out an investigation, where they examined healthy Turkish children aged between 3 and 17 years and presented the same research results: the thickiest quadrant was inferior and the thinnest one was temporal [26]. El-Dairi et al. indicated that RNFL is race dependent and black children have higher RNFL thickness values than white children, especially their superior quadrant was greater [27] (changes are made in lines 189-197). However, F. Zheng et al. in their study determined that there are no meaningful correlations between FD% and axial length (|r| < 0.30) in 20-80 years old healthy adults [29] (changes are made in lines 209-210).
As limitation of our work we pointed that we could not consider diurnal variations of CC perfusion [32]. We found that Tan et al. studied how circadian rhythms affect subfoveal choroidal thickness and they found that the highest values were in the morning at 9:00 and the lowest values were in the evening 5:00 [33] (changes are made in lines 226-229).
Round 2
Reviewer 1 Report
The authors considered all mentioned Comments/Suggestions.
However, in line 93 the automatic segmentation is mentioned. I recommend to add the exact segmentation of the choriocapillaris (CC) slab. I believe the Topcon device defines the CC slab as 0 - 10.4 µm below Bruch's membran. But the authors should check this automated segmentation and report the used CC slab. Furthermore, "JCW" is mentioned as the grader for the OCTA images. However, JCW is neither among the authors, nor mentioned in Contributions section.
Author Response
Point 1: However, in line 93 the automatic segmentation is mentioned. I recommend to add the exact segmentation of the choriocapillaris (CC) slab. I believe the Topcon device defines the CC slab as 0 - 10.4 µm below Bruch's membran. But the authors should check this automated segmentation and report the used CC slab. Furthermore, "JCW" is mentioned as the grader for the OCTA images. However, JCW is neither among the authors, nor mentioned in Contributions section.
Response 1: Thank you for your comment. We deleted about experienced grader (JCW) and added, that Topcon device defines the CC slab as 0 - 10.4 µm below Bruch's membrane. Changes are made in lines 93-96.
Reviewer 2 Report
The manuscript was improved according to the reviewers' comments. I recommend the authors to add the standard deviations in the bar graph of Figure 3.
Author Response
Point 1: The manuscript was improved according to the reviewers' comments. I recommend the authors to add the standard deviations in the bar graph of Figure 3.
Response 1: Thank you for your comment. We added the standard deviations in the bar graph of Figure 3. Changes are made in lines 164-165.